# Trastuzumab-Targeted Biodegradable Nanoparticles for Enhanced Delivery of Dasatinib in HER2+ Metastasic Breast Cancer

**DOI:** 10.3390/nano9121793

**Published:** 2019-12-16

**Authors:** Enrique Niza, María del Mar Noblejas-López, Iván Bravo, Cristina Nieto-Jiménez, José Antonio Castro-Osma, Jesús Canales-Vázquez, Agustín Lara-Sanchez, Eva M. Galán Moya, Miguel Burgos, Alberto Ocaña, Carlos Alonso-Moreno

**Affiliations:** 1Dpto. Inorgánica, Orgánica y Bioquímica, Facultad de Farmacia de Albacete, UCLM, 02071 Albacete, Spain; enrique.niza@uclm.es (E.N.); joseAntonio.Castro@uclm.es (J.A.C.-O.); 2Oncología Traslacional, Unidad de Investigación del Complejo Hospitalario Universitario de Albacete, 02071 Albacete, Spain; marnoblejas22@gmail.com (M.d.M.N.-L.); cnjimenez92@gmail.com (C.N.-J.); mburgoslozano@sescam.jccm.es (M.B.); 3Centro Regional de Investigaciones Biomédicas (CRIB), UCLM, 02008 Albacete, Spain; EvaMaria.Galan@uclm.es; 4Dpto. Química Física, Facultad de Farmacia de Albacete, UCLM, 02071 Albacete, Spain; ivan.bravo@uclm.es; 5Instituto de Energías Renovables, UCLM, 02071 Albacete, Spain; jesus.canales@uclm.es; 6Dpto. Inorgánica, Orgánica y Bioquímica, Facultad de Ciencias y Tecnologías Químicas de Ciudad Real, UCLM, 13075 Ciudad Real, Spain; Agustin.lara@uclm.es; 7Experimental Therapeutics Unit, Hospital clínico San Carlos, IdISSC and CIBERONC, 28029 Madrid, Spain

**Keywords:** Dasatinib, trastuzumab, HER metastasic breast cancer, antibody-targeted nanoparticles

## Abstract

Dasatinib (DAS) is a multikinase inhibitor that acts on several signaling kinases. DAS is used as a second-line treatment for chronic accelerated myeloid and Philadelphia chromosome-positive acute lymphoblastic leukemia. The therapeutic potential of DAS in other solid tumours is under evaluation. As for many other compounds, an improvement in their pharmacokinetic and delivery properties would potential augment the efficacy. Antibody-targeted biodegradable nanoparticles can be useful in targeted cancer therapy. DAS has shown activity in human epidermal growth factor receptor 2 (HER2) positive tumors, so conjugation of this compound with the anti-HER2 antibody trastuzumab (TAB) with the use of nanocarriers could improve its efficacy. TAB-targeted DAS-loaded nanoparticles were generated by nanotechnology. The guided nanocarriers enhanced in vitro cytotoxicity of DAS against HER2 human breast cancer cell lines. Cellular mechanistic, release studies and nanoparticles stability were undertaken to provide evidences for positioning DAS-loaded TAB-targeted nanoparticles as a potential strategy for further development in HER2-overexpressing breast cancer therapy.

## 1. Introduction

The discovery of new targeted therapies is a main objective, particularly for diseases like cancer where mortality is observed in a high proportion of patients [1,2,3]. Protein kinases play a central role in the activation of oncogenic signaling pathways and some small kinase inhibitors can interfere with their activity. As an example of this family of agents, is the multikinase inhibitor dasatinib (DAS) which inhibits, among others, the kinase SRC [4,5,6]. Currently, it is effectively used as a second line treatment for chronic accelerated myeloid and Philadelphia chromosome-positive acute lymphoblastic leukemia. However, the potential role of SRC in other tumor types has been described [7,8,9]. For instance, SRC has been implicated in the development of resistance to the anti-human epidermal growth factor receptor 2 (HER2) antibody trastuzumab (TAB) in HER2 positive breast cancer. In this context, inhibition of SRC with DAS has shown preclinical and clinical activity [10,11,12,13].

Drug-delivery systems (DDS) could be used to enhance DAS bioavailability, minimize adverse side effects and prolong pharmacological activity. For this purpose, poly(ethylene glycol)-poly(ε-caprolactone) copolymer (PEG-PCL) micelles were successfully reported to enhance DAS solubility and improve inhibition of pathologic cellular processes of the retinal pigment epithelium [14]. In another approach, DAS-loaded albumin nanoparticles exhibited potent anti-leukemia efficacy when compared to free DAS treatment [15]. In an elegant study, multifunctional micellar nanoparticles were developed to improve tumour specificity and activity of DAS on lung, cervical, breast and ovarian cancer, and in murine melanoma cells [16]. Magnetically guided micelles were reported as nanocarriers for enhanced delivery of DAS to human triple-negative breast cancer cells [17]. Recently, our group has reported new polymeric nanoparticles (NPs) for controlled DAS delivery in breast cancer therapy [18]. NPs can have several advantages: (1) to prolong the stability in circulation, (2) to increase the capacity to carry high toxic drugs with a very narrow therapeutic margin, (3) to target cancer cells, (4) and to circumvent drug-resistance mechanisms.

Targeting cell receptors is an attractive approach for the treatment of cancer. The possibility to carry targeted therapies in nanocarriers through a conjugation with antibodies is being proposed as a novel strategy in oncology [19,20]. The antibody-drug conjugated nanoparticles (ACNPs) could deliver the drug in a controlled manner, reducing toxicity [21]. The identification of membrane proteins overexpressed in tumoral cells is needed when designing the use of antibodies to be used as the NP vector. TAB is a monoclonal antibody against HER2 positive breast cancer that could be used to vectorize the NPs.

The main objective of this work was to generate a formulation that could improve the therapeutic efficacy of the anti-SRC kinase inhibitor DAS in HER2 metastatic breast cancer. A novel DDS was developed based on encapsulating DAS into biodegradable NPs conjugated with anti-HER2 antibody TAB. The NPs were characterized from particle size, shape and release profiles. The anticancer effect of TAB-targeted DAS-loaded NPs was studied in HER2 overexpressing breast cancer cells. Finally, biochemical studies explored the mechanism of action.

## 2. Materials and Methods 

### 2.1. Materials 

The synthesis of poly-L-lactide (22,000 Da) (PLA) was performed under nitrogen, using standard Schlenk techniques [22]. ZnEt_2_ 1M solution (Sigma-Aldrich, Madrid, Spain) was used as received. L-LA (Sigma-Aldrich) was sublimed three times and stored in the glovebox. Zinc compounds were prepared according to literature procedures [23]. Dasatinib (DAS) were purchased as dasatinib monohydrate (high-performance liquid chromatography (HPLC): 99.60% purity) by Selleckchem (Houston, TX, USA) and trastuzumab (TAB) were purchased as Herceptin by ROCHE (F. Hoffmann-La Roche Ltd).

### 2.2. Characterization

^1^H nuclear magnetic resonance (NMR) spectra were recorded on a Varian Inova FT-500 spectrometer. Gel permeation chromatography (GPC) measurements were performed on a Polymer Laboratories PL-GPC-220 instrument equipped with a TSK-GEL G3000H column and an ELSD-LTII light-scattering detector. Field Emission Scanning Electron Microscopy (FE-SEM) images were recorded on a Jeol 7800 F electron microscope to study the particle size distribution and morphology of the nanoparticles. High-resolution electron microscope images were obtained on a Jeol JEM 2100 transmission electron microscope (TEM) operating at 200 kV and equipped with an Oxford Link EDS detector. As the specimens could be damaged under beam irradiation, observation was performed under low-dose conditions. The resulting images were analyzed using Digital Micrograph™ software from Gatan. The average sizes, polydispersities and Z-potentials of the formulations were measured using a Zetasizer Nano ZS (Malvern Instruments). Data were analyzed using the multimodal number distribution software included in the instrument.

Loading efficiency (LE) and encapsulation efficiency (EE) of DAS were calculated according to the following equations:

LE% = (weight of encapsulated DAS (mg))/(weight of total (DAS encapsulated+scaffold weight) (mg)) × 100%

EE% = (weight of encapsulated DAS (mg))/(weight of DAS feeding (mg)) × 100%

### 2.3. Preparation of Nanoparticles (NPs)

**Polylactide NPs.** The NPs were prepared by nanoprecipitation and displacement solvent method [24]. Briefly, 20 mg of PLA in 3 mL of tetrahydrofuran (THF) was added dropwise into 17 mL of polyvinyl alcohol (PVA) (0.2% aqueous solution) under vigorous stirring. The THF was evaporated under reduced pressure. After centrifugation at 14,000 rpm for 40 min the NPs were collected.

**Polyethyleneimine (PEI) coating NPs.** 20 mg of PLA in 3 mL of THF was added dropwise into a 17 mL-aqueous phase containing 0.5% *w*/*w* of PEI and 0.2% of PVA. The THF was evaporated under reduced pressure. The particle suspension was centrifuged at 14,000 rpm for 40 min at 4 °C to collect the NPs. The suspension was separated into two Eppendorf, one of them with 1 mL of phosphate-buffered saline (PBS) pH 7.4 and another with 1 mL PBS pH 5.8 for subsequent conjugation with TAB.

**DAS-loaded NPs.** The DAS-loaded NPs were prepared by the same methodology described above. Briefly, 20 mg of PLA in 3 mL of THF and 3 mg of DAS in 50 µL of DMSO were mixed to form the organic phase. The organic phase was subsequently added dropwise into 17 mL of PVA (0.2% *w*/*w*) aqueous solution under vigorous stirring. The THF was then evaporated under reduced pressure. The particle suspension was centrifuged at 14,000 rpm for 40 min at 4 °C to collect the NPs.

**DAS-loaded PEI coating NPs.** The DAS-loaded PEI coating NPs were prepared by the same method described above. Briefly, the organic phase was added dropwise into aqueous phase, with 0.5% PEI in 17 mL of PVA 0.2% solution under vigorous stirring. The THF was then evaporated under reduced pressure. The particle suspension was centrifuged at 14,000 rpm for 40 min at 4 °C to collect the NPs. The suspension was separated into two Eppendorf tubes, one of them with 1 mL of PBS pH 7.4 and another with 1 mL PBS pH 5.8 for subsequent conjugation with TAB.

**TAB-conjugated NPs.** The TAB was chemically conjugated to PEI coating NPs after activating [25]. Briefly, 40 mg of 1-ethyl-3-(3-dimethylaminopropyl)carbodiimide (EDC) and 9.7 mg of N-hydroxysuccinimide (NHS) were dissolved in 4 mL of PBS (0.1 M, pH 5.8) followed by the addition of 12 microliters of antiHER2 (21 mg mL^−1^ of TAB in 0.1 M PBS, pH 7.4). 1 mL of PEI coating NPs suspension in PBS pH 5.8 was added to antibody solution and left at room temperature for 12 h. The suspension was centrifuged at 14,000 rpm for 40 min at 4 °C to remove the excess of EDC/NHS. The pellet of TAB conjugated NPs were suspended in PBS pH 7.4. Standard protocol of Bradford assay was employed for quantifying the concentration of the protein in the supernatant.

**DAS-loaded TAB-conjugated NPs.** The NPs were prepared by the same method described above. After TAB activation, 1 mL of DAS-loaded PEI coating NPs suspension in PBS pH 5.8 was added to TAB solution and left at room temperature for 12 h. The suspension was centrifuged at 14,000 rpm for 40 min at 4 °C the pellet suspended in PBS pH 7.4. The standard protocol of Bradford assay was employed for quantifying the concentration of the protein in the supernatant. 

**Stability of NPs in human serum.** The stability of the NPs was performed in 10% serum. Briefly, the NPs were incubated at 37 °C, at concentration equal to 1 mg·mL^−1^. The hydrodynamic radius (R_H_) and polydispersity index (PdI) of the formulations were calculated at predetermined intervals of time by dynamic light-scattering (DLS) measurements.

**Drug-release studies.** To determine the amount of drug release from respective nanoparticles, 10 mg of lyophilized nanoparticles were suspended in 25 mL of phosphate buffered saline (PBS pH 7.4) and sealed in a dialysis membrane (molecular weight cut off: 3500 Da). After incubation at 37 °C, 3 mL of release medium was taken out and replaced by fresh medium at certain intervals. After centrifugation DAS-release concentration was measured in a spectrophotometer at 324 nm. The drug releases were tested in three replicates.

### 2.4. In Vitro Assays

**Cell culture.** HER2+ BT474 and BT474-RH (TAB-resistant) cells and triple negative MDA-MB231 cells were grown in Dulbecco’s modified Eagle medium (DMEM) medium supplemented with 10% inactivated fetal bovine serum. All cell lines used were provided by Drs. J. Losada and A. Balmain, who purchased them from the ATCC, in 2015. Cells authenticity was confirmed by STR analysis at the molecular biology unit at the Salamanca University Hospital. BT474-derived resistant cell line (BT474-RH) was obtained by exposure to trastuzumab (for 6-8 months). All mediums were supplemented with 2 mM L-glutamine, penicillin (20 units/mL) and streptomycin (5 μg/mL). Cells were maintained at 37 °C in a saturated humidity atmosphere with 5% of CO_2_. 

**Toxicity.** Cells were plated onto 48-well plates (10,000 cells/well) and maintained at 37 °C in a saturated humidity atmosphere (5% of CO_2_) until the next day as described previously [24]. HER2+ cells were treated with DAS (50 and 100 nM), TAB (50 nM) and TAB-DAS-(PEI)NPs (50 and 100 nM of DAS) for 72 and 120 h. Also, BT474 were treated with the rest of NPs formulations: NPs, (PEI)NPs, TAB-(PEI)NPs (50 and 100 nM), DAS-NPs and DAS-(PEI)NPs (50 and 100 nM of DAS) for 72 and 120 h. Triple negative cells MDA-MB231 were treated with TAB-DAS-(PEI)NPs, and DAS-(PEI)NPs (50 and 100 nM of DAS) for 72 h. After treatment, 3-(4,5-dimethylthiazol-2-yl)-2,5-diphenyl tetrazolium bromide (MTT) (5 mg/mL) was subsequently added to each well and the cells were incubated at 37 °C for 1 h. The culture medium was removed, and the insoluble formazan crystals were dissolved in 200 μL DMSO (Merck Millipore, Spain). Absorbance (A555 nm–A690 nm) was measured in a multiwell plate reader (BMG labtech, Ortenberg, Germany).

**Drug to antibody ratio study.** For evaluated if TAB cargo has influence in antiproliferation effect of NPs MTT assays was performed as described in toxicity section. BT474 were plated as the same form, and later were treated with TAB-DAS-NPs (25, 50, 75, and 100 nM) with 0,8;1,6 and 3,2 nM TAB cargo, respectively. 

**Cell-cycle studies.** BT474 and BT474-RH cells were seeded in p6-well plates (250,000 cells/well) and treated with DAS (100 nM) and TAB-DAS-NPs (100 nM of DAS) for 24 h Non-treated cells were used as control. Cells were collected and fixed with 70% cold ethanol for 30 min at 4 °C. Then, cells were washed with PBS+2%BSA and stained with propidium iodide/RNAse staining solution (Immunostep S.L.). Results were analyzed on FACSCanto II flow cytometer (BD Biosciences). The percentage of cells in each cell-cycle phase was determined by plotting DNA content against cell number using the FACS Diva software. We presented G0/G1 incremented of cells after each condition of treatment, non-treated cells were used as control. 

**Apoptosis.** BT474 and BT474-RH cells were seeded in p6-well plates (250,000 cells/well) and treated with DAS (100 nM) and TAB-DAS-(PEI)NPs (100 nM of DAS) for 72 h. Non-treated cells were used as control. Then, they were collected and stained in the dark with Annexin V-DT-634 (Immunostep S.L.), and propidium iodide (2 mg/mL) at room temperature for 1 h. Cell death was determined using a FACSCanto II flow cytometer (BD Biosciences). We divided population of cells in living cells (Annexin and PI negatives) and death cells (Annexin and/or PI positive). We presented cell death incremented (in U.A.) after the treatment referred to control, non-treated cells. 

**Toxicity in 3D structure.** BT474 and BT474-RH cells (5000 cells) were resuspended in DMEM+2% of Matrigel (Sigma-Aldrich) and seeded on a layer (1 mm) of Matrigel previously added in a p48-multiplate well. After 24 h of incubation, spheres cultures were treated with DAS (100 nM), TAB (50 nM), and TAB-DAS-(PEI)NPs (100 nM of DAS). 3D cultures growth was monitored 72 h after, taking photos with Nikon Eclipse TS100™, objective 10×/0,25 microscope inverted. Spheres diameter was measured by ImageJ software. We represented the sphere size referred to sphere’s diameter score as arbitrary length units. Sphere diameters used non-treated cells as control. 

**Statistical analysis.** Software GraphPad Prism version 5 was used for statistics analysis. Data are expressed as mean ± s.e.m. from at least three independent experiments. A *t*-test for independent samples non-parametric assay (one-tail) or analysis of variance (ANOVA) with Newman–Keuls post-test was used to determine significant statistical differences between different condition of treatments. The level of significance was considered 95%, so p values lower than 0.05 were considered statistically significant: * *p* ≤ 0.05; ** *p* ≤ 0.01 and *** *p* ≤ 0.001. 

## 3. Results 

### 3.1. Dasatinib (DAS)-Loaded Trastuzumab (TAB)-Conjugated NPs Exhibit Controlled Release of DAS with No Significant DAS Burst Release

Figure 1 shows a schematic representation of the NPs formulation. The FDA-approved Polylactide (PLA) and Polyethyleneimine (PEI) were chosen as building blocks for NPs generation. NPs and DAS-loaded NPs (DAS-NPs) were prepared by nanoprecipitation. The surface of NPs was modified with a positively charged polyethyleneimine (PEI) to produce (PEI)NPs and DAS-loaded (PEI)NPs (DAS-(PEI)NPs). The non-loaded and DAS-loaded NPs were conjugated with Trastuzumab (TAB) by covalent coupling via chemical cross-linking to generate to antibody-targeted NPs (TAB-(PEI)NPs) and TAB-targeted DAS-loaded NPs (TAB-DAS-(PEI)NPs), respectively (see Materials and Methods).

Characterization of NPs were carried out by the dynamic light-scattering (DLS) technique, field-emission scanning electron microscopy (FE-SEM) and TEM (Table 1 and Figure 2). DLS studies showed average particle size of the different formulations close to 120 nm, except for DAS-loaded non conjugated and conjugated NPs which were slightly higher. The increase in the average size is expected after PEI modification. [26]. The TAB conjugation was confirmed by the decrease in the surface charge of NPs (Z-potential) to +32 mV (DAS-(PEI)NPs) to +27.7 mV (TAB-DAS-(PEI)NPs). The final particle size of TAB-DAS-(PEI)NPs was 132.1 nm with a polydispersity index (PdI) of 0.189. TEM images show nanoparticles of approximately 120 nm which exhibit a core-shell morphology. Such distribution is consistent with PEI modification which results in a 5 nm shell surrounding the PLA nanoparticles (see Figure 2b). After conjugation with TAB, the surface of the NPs is modified, and the interaction of antibodies can be clearly observed as shown in Figure 2.

Loading (%LE) and encapsulation efficiency (%EE) of DAS-loaded formulations are depicted in Figure 3. TAB-DAS-(PEI)NPs showed high %EE of more than 90% with an active LE of 11.6% *w*/*w*. Higher %LE were obtained when compared to DAS-loaded magnetic micellar nanoparticles [15] and similar to those reported by encapsulating in albumin micelles [17]. In vitro release of DAS-loaded NPs was carried out using the dialysis method at pH 7.4 to mimic the physiological pH of circulation. The release mechanism for polymeric NPs is based on triphasic profiles where a first fast step belongs to “burst release”, followed by a second diffusion step through the pores and channels, to end up with a third deflation lap [27]. As illustrated in Figure 3, TAB-DAS-(PEI)NPs exhibited a controlled release of DAS. TAB-DAS-(PEI)NPs only showed a DAS burst release of less than 15% at pH 7.4; then a sustained drug release profile was achieved in which 60% of DAS was released after 72 h. In case of DAS-NPs, the release was nearly completed (92%) after 72 h at physiological conditions.

### 3.2. DAS-Loaded TAB-Conjugated NPs Display Potent and Selective Cytotoxicity in Breast Cancer Cells

The in vitro cytotoxicity of the formulations was examined by MTT assay in two cell line models, one is BT474, a classic cell line that overexpress HER2, and the second model is BT474–RH, the same cell line but resistant to TAB after being treated with the antibody for a long period of time. The generation of this cell lines was described elsewhere [28]. Both models recapitulate human breast cancer in different clinical situations. BT474 and BT474-RH cells were treated at 50 and 100 nM for 72 and 120 h. Non-loaded NPs (NPs, (PEI)NPs, and TAB-NPs) did not display any significant cytotoxicity in tumoral cells, indicating an appropriate biosecurity profile of the NPs (Figure 4a). An enhanced cytotoxicity of conjugated and non-conjugated DAS-loaded NPs was observed in both cells (Figure 4b). Figure 5 showed an effect of free DAS (IC_50_~100 nM (72 h)), and TAB-DAS-(PEI)NPs (IC_50_~50 nM (72 h)) in BT474 and BT474-RH cancer cells. Finally, the administration of TAB-DAS-(PEI)NPs was more active than administration of single agent TAB or DAS at different time points (72 h and 120 h), indicating the efficacy of the vectorized NPs.

We confirmed the cell viability effect of TAB-DAS-(PEI)NPs in 3D spheroid cultures generated from BT474 and BT474-RH cell lines (Figure 5). 3D spheroid cultures, constitute a more physiologically model than 2D cell cultures for the evaluation of novel therapeutic strategies. As observed for 2D cell cultures, the invasion capacity of matrigel-embedded 3D cultures of BT474 and BT474-RH cells was significantly reduced after TAB-DAS-(PEI)NPs treatment (Figure 6).

Next, we used the non-over expressing HER2 cell line MDAMB-231 to confirm that the effect was secondary to the binding of the NPs to HER2. Administration of TAB-DAS-(PEI)NPs showed similar MTT inhibition to DAS-(PEI)NPs at two different doses 50 nM and 100 nM after 72 h (Figure 7a). These results demonstrated that conjugation with TAB facilitates the uptake of nanoparticles by targeting cells that overexpress HER2 and had similar effects in those that do not express HER2. The physical stability of TAB-DAS-(PEI)NPs was studied in PBS at different time points. The values of R_H_ and PdI of the TAB-DAS-(PEI)NPs were measured by DLS (Figure 7b). The negligible increase in either particle size or PdI during a 7-day long experiment suggest high stability against aggregation. The antiproliferative activity of TAB-DAS-(PEI)NPs on BT474 were measured by MTTs assays at different times. The TAB-DAS-(PEI)NPs remained active after 3 months of preparation and storage as NPs suspension at 4 °C (Figure 7c). It is important to note for further clinical development that the lyophilization of TAB-DAS-(PEI)NPs decreased cytotoxicity activity of the formulation (Figure 7d).

Drug to antibody ratio (DAR) can be an important factor influencing effectiveness of Antibody drugs Conjugate NPs (ACNPs) [29]. DAR must be homogenous throughout the NPs and with an optimal balance between cytotoxicity and pharmacokinetic profile. Figure 8 showed MTTs assays of TAB-DAS-(PEI)NPs with different TAB cargo (0,8 nM to 3,2 nM). No significant differences based on different cargoes were observed at several concentrations of TAB-DAS-(PEI)NPs on the cytotoxicity of BT474 cells.

### 3.3. DAS-Loaded TAB-Conjugated NPs Increment Cycle Arrest and Cell Death More in TAB-Resistant Cells

To explore whether their mechanism of action was different from free DAS, the two HER2+ cell lines, BT474 and BT474-RH, were treated with TAB-DAS-(PEI)NPs and stained with propidium iodide/RNase solution at 24 h of treatment. DAS blocked the progression through the G1/G0-phase boundary. Administration of TAB-DAS-(PEI)NPs showed a slight increase in G1 compared to the free drug, what confirmed that the NPs mediates their effect in the same manner as DAS in its free formulation (Figure 9a). On the other hand, Figure 9b showed enhanced apoptosis in resistant cells, treated with TAB-DAS-(PEI)NPs in comparison with free DAS and free TAB, which suggests that the resistant cells rely more on this kinase that the naïve ones, a finding in line with previous reports [12].

## 4. Discussion

Breast cancer remains one of the most common malignancies worldwide and the HER2 positive breast cancer subtype constitutes 25% of this population. Despite the development and implementation of new treatments to our daily clinical armamentarium, HER2 positive metastatic breast cancer remains an incurable condition. In this context, the discovery, design and optimization of novel and improved therapeutic strategies is a main objective. Antibody-drug conjugated nanoparticles (ACNPs) represent a relatively new approach that is based on the success and potential of antibody conjugation and nanotechnology [19,20]. In comparison with antibody-drug conjugates (ADC), ACNPs can deliver the drug in a controlled manner preserving its chemical structure, avoiding unpredicted metabolization, and reducing toxicity. The combination of chemotherapies in NPs offers the opportunity to overcome pharmacokinetic differences in drug agents to ensure their delivery at the disease site in the required proportions. The diversity of drug agents that can be incorporated into ACNPs offers further development opportunities than can be afforded with standard ADC technologies. In this context, the main objective of this work was to develop ACNPs for the treatment of breast cancer. DAS was encapsulated in trastuzumab-vectorized NPs with the objective to improve its solubility and avoid its rapid metabolism. 

The DAS nanocarriers obtained for this purpose were characterized by size, PdI and superficial charge. The average NPs size was approximately 100 nm, in the same range or even smaller than the values reported for DAS encapsulation by other authors in albumin nanoparticles and magnetic micelles [15,17]. Similar %LE was obtained to DAS encapsulation in albumin nanoparticles with slightly smaller average size and Z-potential values [15]. Considerably higher values for LE was obtained in comparison with the polymeric micelles reported for the DAS encapsulation [14]. Once the formulation was optimized reaching an EE value close to 90%, TAB were attached over the surface after PEI coating and covalent binding by EDC/NHS chemistry [26]. PEI coating for antibody conjugation was chosen for this first approach due to its easier formulation and low cost. In any case, ACNPs showed Z-potential to guarantee enough stability for further in vitro studies. 

In vitro release studies of the ACNPs showed a sustained release of DAS over 72 h with a negligible burst release compared to non-triggered DAS-loaded NPs. The DAS release from the polymeric NPs was comparable to that reported from polymeric micelles and metal nanoparticles [14,15,16,17]. After PEI coating and subsequent antibody conjugation the nanocarriers achieved more sustained DAS release over time. 

ACNPs induced cytotoxicity specifically in cell lines overexpressing HER2 like BT474, showing more activity than DAS alone. In TAB-resistant cell lines, BT474-RH, TAB-DAS-(PEI)NPs showed more activity than single agents alone, demonstrating the vulnerability that constitute SRC inhibition on this cell population. Non-loadable NPs did not show activity confirming the security profile of the nanocarriers. Non-targeted nanovehicles were also assessed on cells lacking the expression of HER2, to confirm that the specificity of the effect depended on the binding with TAB to the receptor. As a result, it was shown that ACNPs were much more efficient than free DAS and non-targeted DAS-loaded nanovehicles. ACNPs reduced non-HER2-overexpresing cell lines availably in the same way than non-targeted DAS nanocarriers which demonstrates the specificity of the strategy. 

Cellular mechanistic studies of the DAS-loaded ACNPs confirmed the induction of apoptosis in HER2 overexpression breast cancer cells, as well as cell-cycle arrest in the G1/G0-phase. Of note, the effect was similar in both BT-474 and BT-474RH cells although the activity was more pronounced in the latter, suggesting that the mechanism of action was similar, but the resistant clones depended more on SRC inhibition. We are aware that an in-depth evaluation of the binding and internalization process would provide relevant information. In this context, efficient internalization of ACNPs have a clear role in its efficacy. However, we consider that this kind of study was beyond the aim of this work that mainly focused on the generation and characterization of the NPs. Finally, stability studies regarding storage and drug-antibody ratio of the ACNPs were performed to assess their potential use in further in vitro and in vivo studies. The activity of the ACNPs after storage in suspension at 4 °C was maintained over three months. No significant changes were observed in RH and PdI, indicative of no significant aggregation after being storage. The activity of ACNPs was reduced after lyophilisation, whereas the TAB cargo over the NPs did not influence significantly over the activity of the ACNPs.

## 5. Conclusions 

In this study we demonstrate that the encapsulation of DAS into TAB-targeted biodegradable polymeric NPs resulted in in vitro efficacy, particularly in HER2-overexpressing cells, maintaining the same mechanism of action as DAS given alone. In addition, the generated NPs are stable over time. These results open the door for further assessment of efficacy and safety using in vivo studies that could be the based for its future clinical development.

## Figures and Tables

**Figure 1 nanomaterials-09-01793-f001:**
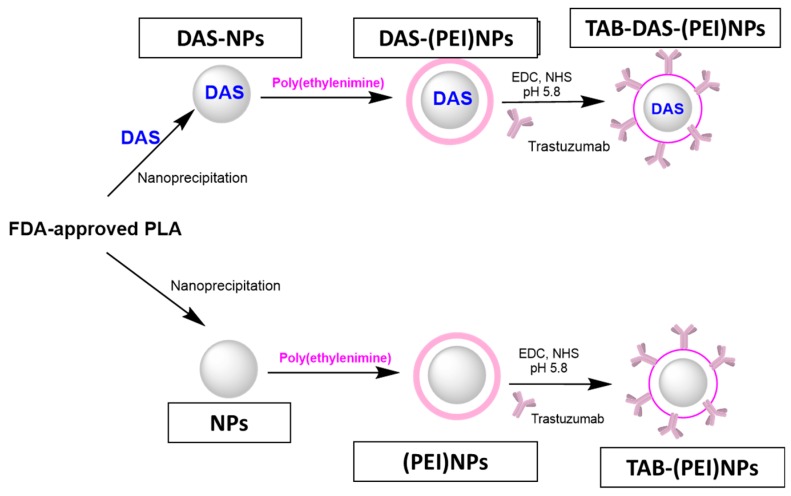
Schematic representation of the nanoparticles’ (NPs) generation.

**Figure 2 nanomaterials-09-01793-f002:**
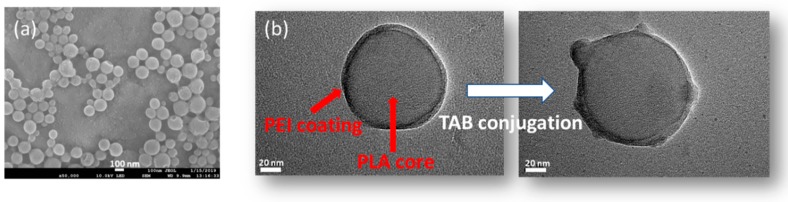
Antibody conjugation is illustrated by field-emission scanning electron microscopy (FE-SEM) and transmission electron micrsocopy (TEM) images. (**a**) FE-SEM image of trastuzumab- dasatinib coated nanoparticles (TAB-DAS-NPs) (**b**) TEM images of polyethyleneimine-coated nanoparticles (PEI)NPs before (left) and after (right) TAB conjugation.

**Figure 3 nanomaterials-09-01793-f003:**
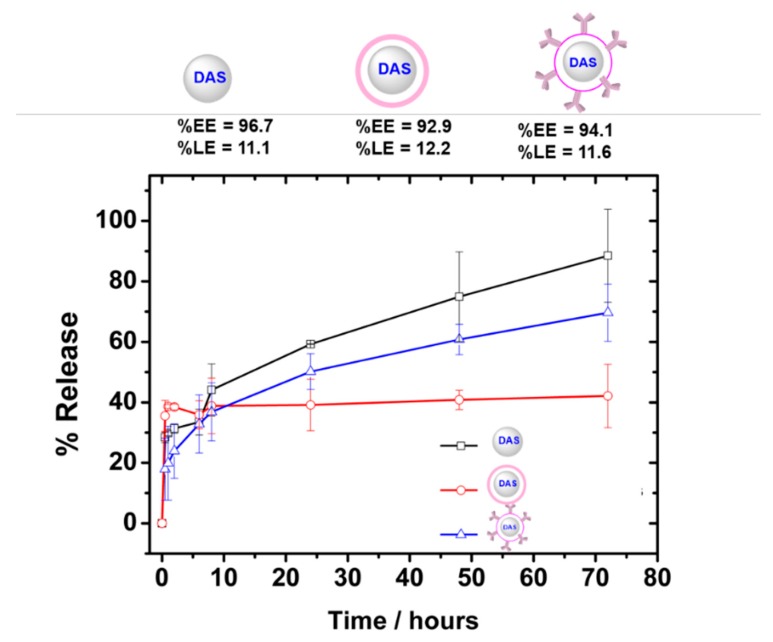
TAB-DAS-NPs shows sustained DAS release with negligible burst release. Efficiency encapsulation, loading efficiency and in vitro release profiles in phosphate buffered saline (pH 7.4) at 38 °C of DAS-loaded NPs formulations.

**Figure 4 nanomaterials-09-01793-f004:**
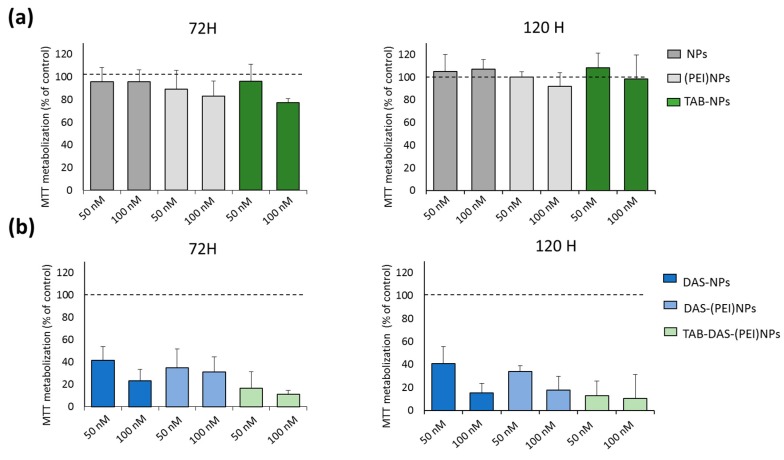
(**a**) Non-loaded NPs did not show antiproliferative effect in BT474. (**b**) DAS-loaded NPs exercised antiproliferative effect in BT474. Metabolization of MTT was determined by spectrophotometry. Results was referred to control (non-treated cells) as 100% of MTT metabolization.

**Figure 5 nanomaterials-09-01793-f005:**
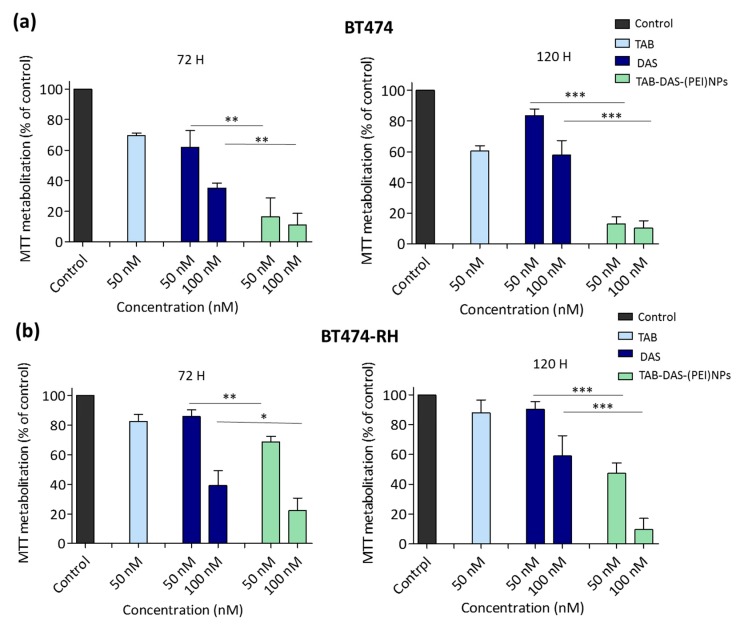
TAB-DAS-NPs exercised antiproliferative effect in TAB sensitive and resistant HER2+ cells. BT474 (**a**) and BT474- RH (**b**) were exposed to DAS, TAB, and TAB-DAS-NPs for 72 and 120 h at the indicated concentrations. Metabolization of MTT was determined by spectrophotometry. * *p* < 0.05; ** *p* < 0.01; *** *p* < 0.001.

**Figure 6 nanomaterials-09-01793-f006:**
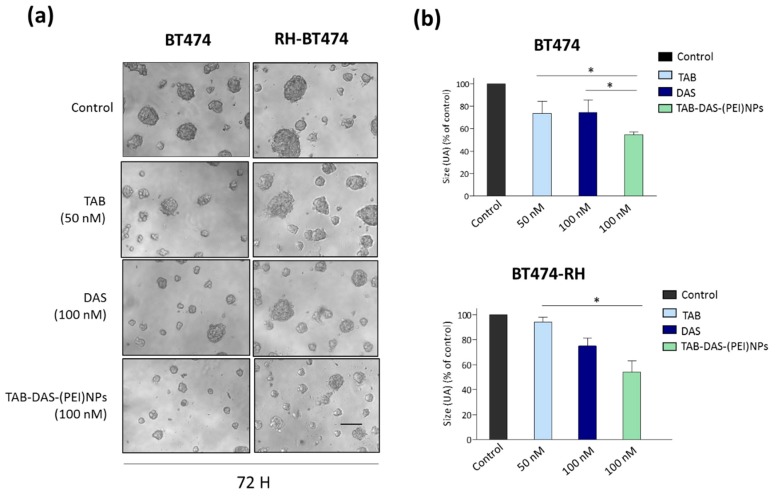
Invasion capacity of matrigel-embedded 3D cultures of BT474 and BT474-RH cells is reduced with TAB-DAS-NPs. Cells were grown in a semi-solid matrigel matrix. Then, 3D cultures were exposed to the indicated doses of the drugs. After 72 h was taken pictures (**a**) and quantified the spheres size (**b**). Scale bar= 100 μm. * *p* < 0.5; ** *p* < 0.005; *** *p* < 0.001.

**Figure 7 nanomaterials-09-01793-f007:**
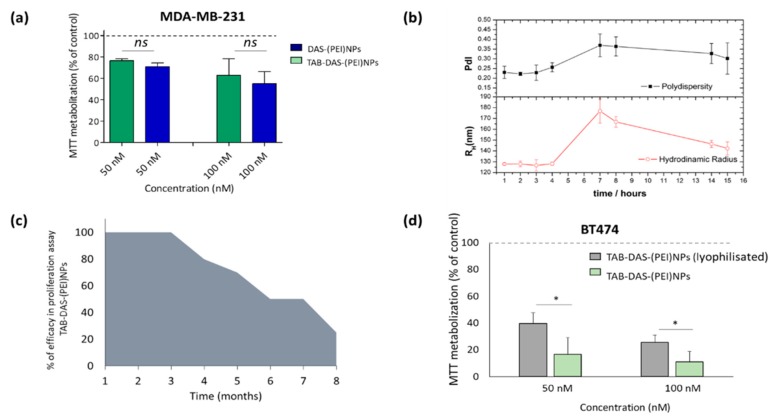
The physical stability of TAB-DAS-(PEI)NPs suitable for further translational studies. (**a**) Antiproliferative activity of TAB-DAS-NPs on HER2 non-over expressing cell line MDAMB-231 after 72 h. (**b**) R_H_ and PdI evolution of TAB-DAS-(PEI)NPs after storage. (**c**) Antiproliferative activity of TAB-DAS-(PEI)NPs on BT474 at different times after storage. (**d**) Antiproliferative activity of TAB-DAS-(PEI)NPs on BT474 after liophylisation. * *p* < 0.5; ** *p* < 0.005; *** *p* < 0.001. *NS* = no significant difference. (**a**) and (**d**) results was referred to control (non-treated cells) as 100% of MTT metabolization.

**Figure 8 nanomaterials-09-01793-f008:**
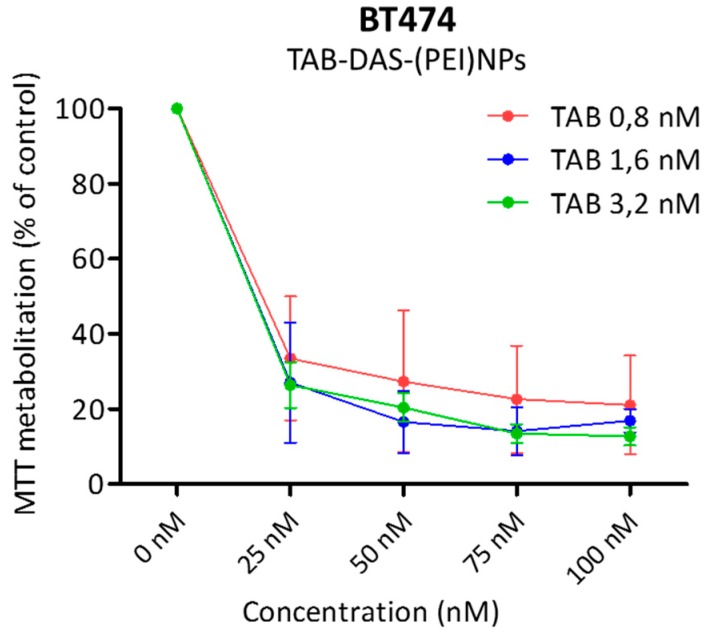
DAR indicates no significant differences of the different cargo over the NPs formulation. BT474 cells were treated with indicated dosis of TAB-DAS-(PEI)NPs with different TAB cargoes (indicated in legend). MTT assays was performed for analysed antiproliferative effect.

**Figure 9 nanomaterials-09-01793-f009:**
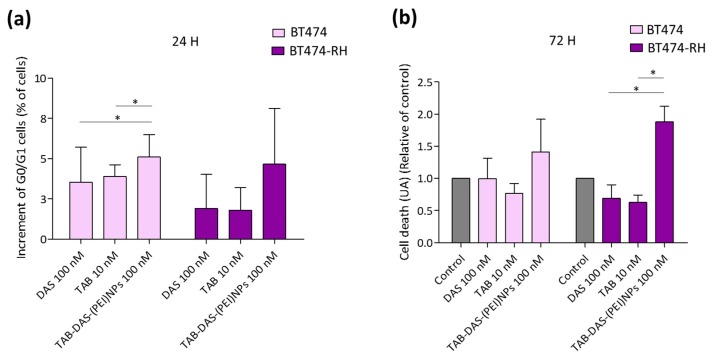
TAB-DAS-(PEI)NPs increment cycle arrest and cell death more in TAB resistant cells. BT474 and BT474-RH were exposed to free DAS, free TAB, and DAS-loaded formulations at the indicated concentration. (**a**) Parental and TAB-resistant BT474 cells increment of G0/G1 after 24 h of treatment compared to control. (**b**) Cell death presented in U.A. relatives of non-treated control cells after 72 h of treatments. * *p* < 0.05; ** *p* < 0.01; *** *p* < 0.001.

**Table 1 nanomaterials-09-01793-t001:** Hydrodynamic diameter (nm), polydispersity index (PdI) and Z-potential of the different formulations obtained by dynamic light-scattering (DLS) measurements.

NPs-Formulation	Average Size (nm)	PdI	Z-potential (mV)
NPs	115.0 ± 0.4	0.41 ± 0.03	−14.4 ± 1.3
DAS-NPs	119.3 ± 1.0	0.23 ± 0.01	−12.2 ± 1.2
(PEI)NPs	120.8 ± 0.9	0.16 ± 0.05	+55.7 ± 1.5
DAS-(PEI)NPs	155.5 ± 2.6	0.52 ± 0.11	+32.1 ± 0.6
TAB-(PEI)NPs	115.5 ± 1.8	0.21 ± 0.02	+46.3 ± 1.0
TAB-DAS-(PEI)NPs	132.1 ± 2.5	0.19 ± 0.01	+27.7 ± 0.5

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
