# Peer review of "Trastuzumab-Targeted Biodegradable Nanoparticles for Enhanced Delivery of Dasatinib in HER2+ Metastasic Breast Cancer"

_nanomaterials, 2019, doi:10.3390/nano9121793_

Round 1

Reviewer 1 Report

This manuscript describes the development of biodegradable nanoparticle with specificity for HER2 based on Trastuzumab and FDA-approved polylactide (LA). The resultant nanoparticle showed cytotoxicity for HER2-positive cell lines. This manuscript has a certain novelty and is of interest to the readers; however, the manuscript would be more improved if the authors revised according to the following comments.

There are no discussion about comparing with other Dasatinib or drug-loaded nanoparticles with/without specificity for target molecules on cytotoxicity.

There are no data in the case of other antibodies. This method can apply to other antibodies? The nanoparticle with irrelevant antibodies actually do not induce cytotoxicity for HER2-positive cell lines?

What is the mechanism of releasing DAS at pH 7.4?

Is PEI also available in clinical studies?

In Fig. 3, why did DAS-(PEI)NPs show lower releasing of DAS than TAB-DAS-(PEI)NPs and DAS-NPs?

Which is correct that MDAMB-231 is non-over expressing HER2 cell line (line 292) or HER2 nonexpressing cell line (line 309)?

Fig. 7 (but described as Figure 6!) showed no differences between TAB-DAS-(PEI)NPs and DAS-(PEI)NPs for HER2 nonexpressing MDAMB-231. But, in Fig. 4b, there are also no major differences among DAS-NPs, TAB-DAS-(PEI)NPs, and DAS-(PEI)NPs for HER2-positive cell line.

In Fig.8, why are there no antibody concentration dependency?

Author Response

 This manuscript describes the development of biodegradable nanoparticle with specificity for HER2 based on Trastuzumab and FDA-approved polylactide (LA). The resultant nanoparticle showed cytotoxicity for HER2-positive cell lines. This manuscript has a certain novelty and is of interest to the readers; however, the manuscript would be more improved if the authors revised according to the following comments.

There are no discussion about comparing with other Dasatinib or drug-loaded nanoparticles with/without specificity for target molecules on cytotoxicity.

Response. This is an interesting observation. However, we consider that we have extensively discussed in the discussion section the available data regarding works that have encapsulated dasatinib with different formulations. 

There are no data in the case of other antibodies. This method can apply to other antibodies? The nanoparticle with irrelevant antibodies actually do not induce cytotoxicity for HER2-positive cell lines?

Response. We consider that our findings could apply to other antibodies. However, in the case of HER2 only trastuzumab has reached the clinical setting. Probably, a non targeting HER2 antibody would not have antitumoral effect.  

What is the mechanism of releasing DAS at pH 7.4?

Response. As it is very well-known for polymeric nanoparticles the working mechanism is based on triphasic profiles where a first fast step belongs to “burst release”, followed by a second diffusion step through the pores and channels, to end up with a third deflation lap. Our  experiment in PBS (pH 7.4) showed negligible burst release. The deflation lap was not observed during the experiment.

Is PEI also available in clinical studies?

Response. Polyethyleneimine is biocompatible, biodegradable and FDA approved agent for human medical applications.

In Fig. 3, why did DAS-(PEI)NPs show lower releasing of DAS than TAB-DAS-(PEI)NPs and DAS-NPs?

Response. The activation of the PEI-coating NPs before conjugation must decrease the thickness of the surface which could explain the reason because TAB-DAS(PEI)NPs and DAS-NPs present similar release profiles.  

Which is correct that MDAMB-231 is non-over expressing HER2 cell line (line 292) or HER2 nonexpressing cell line (line 309)?

Response. MDA MB231 is a triple negative breast cancer cell that express low levels of HER2, so is a non over expressing cell line.

Fig. 7 (but described as Figure 6!) showed no differences between TAB-DAS-(PEI)NPs and DAS-(PEI)NPs for HER2 nonexpressing MDAMB-231. But, in Fig. 4b, there are also no major differences among DAS-NPs, TAB-DAS-(PEI)NPs, and DAS-(PEI)NPs for HER2-positive cell line.

Response. This is correct. This fact supports that the antibodies were used only for the vectorization of the nanoparticles. The conjugated NPs are not active against the non-over expressing HER2 cell line MDAMB-231 (Figure 7) and the non-loaded conjugated NPs are not either (Figura 4b).

In Fig.8, why are there no antibody concentration dependency? 

Response. There is saturation of the receptor when doses of trastuzumab reach more than 50nM.

Reviewer 2 Report

The paper I reviewed is titled: Trastuzumab-targeted biodegradable nanoparticles for enhanced delivery of Dasatinib in HER2+ metastasic breast cancer. The paper is interesting, well written and relevant to the field. The content is original and worth of publication. The improvement of the present paper with respect to other published papers is the formulation of the nanoparticles. In particular the combination of Dasatinib and Trastuzumab in the same formilation. I suggest to accept the manuscript after minor revision upon corrections to minor check of English spell, typos and text editing.

I suggest to check the manuscript for minor English improvements and there are some typos here and there.

Author Response

The manuscript has been checked to correct typos. We really appreciate the revision done and the comments from the reviewer

Round 2

Reviewer 1 Report

Response. As it is very well-known for polymeric nanoparticles the working mechanism is based on triphasic profiles where a first fast step belongs to “burst release”, followed by a second diffusion step through the pores and channels, to end up with a third deflation lap. Our  experiment in PBS (pH 7.4) showed negligible burst release. The deflation lap was not observed during the experiment.

Even if well-known mechanism on nanoparticles, the authors should describe the mechanism of releasing DAS briefly with appropriate citations

Response. Polyethyleneimine is biocompatible, biodegradable and FDA approved agent for human medical applications.

The authors should describe that Polyethyleneimine is also FDA approval as well as Polylactide.

Response. MDA MB231 is a triple negative breast cancer cell that express low levels of HER2, so is a non over expressing cell line.

“non-over expressing” and “nonexpressing” have different meaning each other. Reword nonexpressing as non-over expressing.

Response. This is correct. This fact supports that the antibodies were used only for the vectorization of the nanoparticles. The conjugated NPs are not active against the non-over expressing HER2 cell line MDAMB-231 (Figure 7) and the non-loaded conjugated NPs are not either (Figura 4b).

You mean there are no data on the showing superiority of TAB-DAS-(PEI)NPs to DAS-(PEI)NPs?

Response. There is saturation of the receptor when doses of trastuzumab reach more than 50nM.

Why did you not perform this experiment under lower concentration of TAB-DAS-(PEI)NPs or TAB? The result of TAB 0 nM is also needed. I could not catch what fact you wanted to show using Figure 8.

Author Response

Even if well-known mechanism on nanoparticles, the authors should describe the mechanism of releasing DAS briefly with appropriate citations

Response2. The mechanism of release is included in the revised manuscript with proper references.

The authors should describe that Polyethyleneimine is also FDA approval as well as Polylactide.

Response2. PEI is included in the revised manuscript as FDA approval polymer as well.

“non-over expressing” and “nonexpressing” have different meaning each other. Reword nonexpressing as non-over expressing.

Response2: Non-over expressing replaces nonexpressing in the revised manuscript.

You mean there are no data on the showing superiority of TAB-DAS-(PEI)NPs to DAS (PEI)NPs?

Response2: Excuse my bad explanation. Yes, there is. It is shown in Figure 4. We attempted to say that the non-loaded NPs after antibody conjugation did not show antiproliferative effect in BT474 (Figure 4a). In this context, the TAB-DAS-(PEI)NPs and DAS-(PEI)NPs showed similar inhibition on the non-over expressing HER2 cell line MDAMB-231 (Figure 7a). Consideration these results, we wanted to demonstrate that the conjugation with TAB facilities the uptake by targeting cells that overexpresss HER2.

Why did you not perform this experiment under lower concentration of TAB-DAS-(PEI)NPs or TAB? The result of TAB 0 nM is also needed. I could not catch what fact you wanted to show using Figure 8.

Response2. Previously in Figure 7b we showed the differences between TAB-DAS-(PEI)NPs (TAB 0,8nM) and DAS-(PEI)NPs (TAB 0nM).  0,8nM of TAB is the less quantity of TAB we have managed to conjugate over our NPs. With Figure 8 we want to show that no significant differences based on different cargo were observed at different concentrations. We think that an increase amount of trastuzumab does not increase activity as the saturation of the receptors is achieved and translates into efficacy. We do consider that figure 8 is relevant in terms of drug development.